Global vision object detection using an improved Gaussian Mixture model based on contour

Sun Lei sunly029@foxmail.com
School of Information Engineering, Suqian University , Suqian , Jiangsu , China
Sun Gengxin
Electronic publication date: 2024 Jan 18
Publication date: 2024
Volume: 10
Electronic Location ID: e1812
Received 2023 Aug 23; Accepted 2023 Dec 18
Copyright: ©2024 Sun
Copyright year: 2024
Copyright holder: Sun
License: This is an open access article distributed under the terms of the Creative Commons Attribution License, which permits unrestricted use, distribution, reproduction and adaptation in any medium and for any purpose provided that it is properly attributed. For attribution, the original author(s), title, publication source (PeerJ Computer Science) and either DOI or URL of the article must be cited.
License URL: https://creativecommons.org/licenses/by/4.0/

Keywords: Object detection, Improved gaussian mixture model, Otsu method, Features fusion

Funding: Suqian Sci&Tech Program Z2022102 K202348 Suqian University Scientific Research Fund for Talent Introduction 2023XRC004 The research work described in this paper is supported by Suqian Sci&Tech Program (Grant No. Z2022102 and No. K202348) and Suqian University Scientific Research Fund for Talent Introduction (Grant No. 2023XRC004). The funders had no role in study design, data collection and analysis, decision to publish, or preparation of the manuscript.

==============================
Object detection plays an important role in the field of computer vision. The purpose of object detection is to identify the objects of interest in the image and determine their categories and positions. Object detection has many important applications in various fields. This article addresses the problems of unclear foreground contour in moving object detection and excessive noise points in the global vision, proposing an improved Gaussian mixture model for feature fusion. First, the RGB image was converted into the HSV space, and a mixed Gaussian background model was established. Next, the object area was obtained through background subtraction, residual interference in the foreground was removed using the median filtering method, and morphological processing was performed. Then, an improved Canny algorithm using an automatic threshold from the Otsu method was used to extract the overall object contour. Finally, feature fusion of edge contours and the foreground area was performed to obtain the final object contour. The experimental results show that this method improves the accuracy of the object contour and reduces noise in the object.

Introduction

Visual object detection plays an important role in the field of computer vision. It has a variety of significant applications in human–computer interfaces, road traffic control, and video surveillance systems (Ingle & Kim, 2022; Roy, 2017; Shirpour et al., 2023). In addition to classic applications, an increasing number of scholars are expanding the application of object detection to other fields, such as aquaculture and social computing (Li et al., 2023; Wu et al., 2023). The purpose of object detection is to identify the objects of interest in the image and determine their categories and positions. A higher object detection accuracy corresponds to a higher subsequent recognition and segmentation performance (Liu et al., 2023). However, object detection remains a challenging problem because the background and object of the image can be extremely complex and variable, especially in outdoor environments. Problems encountered by object detection include illumination changes, presence of background clutter, scale changes, and rotation. To manage these problems, the three most commonly used moving object detection methods are: the optical flow method, inter frame difference method, and background subtraction method.

Optical flow is an algorithm used to judge whether there is a moving object based on the vector of the optical flow field. It can detect objects in both static and dynamic backgrounds (Douini et al., 2017). The optical flow method is more accurate in object detection, but the calculation of the optical flow field is complex, and this method is prone to noise interference. The most common optical flow methods used for object detection are the H-S method (Ramli et al., 2016) and the L-K method (Patel & Upadhyay, 2013). The inter frame difference method is used to differentiate two or three adjacent and consecutive frames of images to obtain the contour of a moving object (Chi, 2014). It has a low time complexity, and is a type of algorithm frequently used in the object detection field. But when there is significant noise in the video sequences, the algorithm’s detection error significantly increases. When the object indicates a large area of uniform grayscale, the phenomenon of holes occurs, causing the object to be segmented into multiple regions. This method is best suited for scenarios involving moderate object movement speed and high real-time requirements. Background subtraction also uses threshold comparison to determine the target motion region. This method compares the results of the difference between the current frame image and the background image (Panda & Meher, 2015). Its principal concept is similar to the inter frame difference method, but the main difference is that the background image is calculated through models, especially common models such as the Gaussian model and mixed Gaussian model. Background subtraction is more suitable for environments with fixed image sensors. It is characterized by a simple structure, easy implementation, and good results. With the development of deep learning technology, more scholars are paying attention to the application of the convolutional neural network (CNN) in the field of object detection. Girshick was the first to introduce deep learning into object detection through the regional convolutional neural network object detection model (RCNN) (Saddique et al., 2020). Subsequently, Redmon proposed the YOLO framework, which uses a neural network to predict the location and categories of targets within images (Bisht et al., 2022). Although the detection speed reaches 45 fps, the detection accuracy is low. These two frameworks are typical applications of deep learning object detection, but deep learning detection requires a large number of learning samples, therefore there is no way to detect objects of any uncertain types.

So far, there have been no object detection algorithms suitable for every scenario, resulting in a need for the design of more targeted algorithms based on the characteristics of actual scenarios. The Gaussian mixture model (GMM) method is more suitable for moving object detection in static backgrounds. Bariko et al. (2023) proposed a parallelized implementation of the GMM algorithm on the C6678 digital signal processor (DSP) with eight cores. This method solved the problem of high computational demands by three parallel optimizations of the OpenMP program: first, manually optimized approach and second, manually optimized approach. But it does not improve the object detection performance of the algorithm itself. Jin, Niu & Liu (2019) proposed an improved GMM based on the automatic segmentation method to detect floating objects on water surfaces. In this method, a new strategy is used to update the background model in order to segment the objects floating on water surfaces more effectively. The final result of the object contour is obtained through the graph cutting method. Several parameters need to be manually set in the graph cut, and unreasonable parameters can result in the inability to segment the object boundary. Additionally, graph cutting is computationally complex, making it unsuitable for automatic segmentation. Xue & Jiang (2018) applied the GMM model to object recognition and classification based on ultra-wide band (UWB) signals, and analyzed the impact of different weather backgrounds on the results. The analysis of weather’s impact is only applicable to the field of wireless communication and has little significance to the field of video detection. In this article, an improved feature fusion Gaussian mixture model with edge contour is proposed to solve the problems of unclear foreground contour and excessive noise points in the global vision environment for uncertain types. Uncertain type refers to any type and shape of object, such as humans, cars, animals, etc.

Background Subtraction

Single Gaussian model

The single Gaussian model is based on this assumption: The grayscale values of each pixel in a certain image follow Gaussian distribution on the timeline (Dong et al., 2018). The Gaussian distributions of each point do not affect each other. Assuming Xt represents the pixel value of the image at time t, utandΣt are the mean and variance at time t. So the following Eq. (1) can be used to represent a single Gaussian model: (1) ηXt,ut,Σt=12πn2|Σt|12e−12Xt−utTΣt−1Xt−ut

Then update the mean and variance with Eqs. (2) and (3), where ρ is the learning rate:

(2) ut=1−ρ×ut−1+ρ×Xt

(3) σt2=1−ρ×σt−12+ρ×Xt−ut−1T×Xt−ut−1

Gaussian mixture model

In the single Gaussian model, the probability density function obeys normal distribution. This Gaussian model can achieve ideal results in relatively simple environments. If the background image is complex and there is a lot of interference, there are many deviations in the detection results. It is necessary to use multiple Gaussian distributions for object detection. The Gaussian mixture model (GMM) is an extension of the single Gaussian model (Xue & Jiang, 2018). It is the precise quantification of variable distribution using multiple Gaussian probability density functions which decomposes the variable distribution into several statistical models based on the Gaussian probability density function. The GMM resolves the need to update the background model parameters by continuous iteration because it can smoothly approximate the distribution of any shape. The mixed Gaussian model consists of the following steps (Yan et al., 2020):

Establish a background model:

Using K Gaussian distributions as the background, the probability function is shown in Eq. (4): (4) PXi,t+1= ∑i=1kωi,t+1ηXi,t+1;μi,t+1,Σi,t+1

In the above equation, K is the number of Gaussian models, and its value typically ranges from 3 to 5. Xi,t+1 is the value of the pixel at time t+1. The weight, mean, and covariance of this pixel are represented by, ωi,t+1, μi,t+1,  and Σi,t+1, respectively. The Gaussian distribution probability density function η is represented as Eq. (5): (5) ηXi,t+1;μi,t+1,Σi,t+1=12πn/2|Σi,t+1|1/2e−12Xi,t+1−μi,t+1TΣi,t+1−1Xi,t+1−μi,t+1

Model matching: In the generated K single Gaussian models, if the absolute value of the difference between the current pixel value and the mean of a Gaussian distribution is less than the standard deviation σ D times, it can be determined that the two match. D is the deviation threshold, which is one of the basic parameters of the model. It is represented as Eq. (6). (6) xt−ui,t<Dσi,t

Model updating: If the difference between the pixel values at time t and the mean of a certain Gaussian model satisfies Eq. (6), the two values match and the model parameters ωi,t, μi,t, and Σi,t are updated using Eqs. (7), (8), (9) and (10). If the two do not match, a new Gaussian model with a smaller weight, larger variance, and a mean of that pixel value is used to update the model with the lowest priority in the old mixed Gaussian model. The following are the updated formulas:

(7) ωi,t+1=1−αωi,t+αMi,t

(8) ρ=α/ωi,t

(9) μi,t+1=1−ρμi,t+ρXi,t

(10) σi,t+12=1−ρσi,t2+ρXi,t+1−μi,t+1TXi,t+1−μi,t+1

In the above equations, α and ρ are learning rates for weights and means, respectively. If the Gaussian distribution matches, then Mi,t = 1, otherwise Mi,t = 0.

Background model: This step mainly extracts the pixels that best describe the background effect. Sort ωi,tσi,t in descending order and select the first B Gaussian models as the background model. (11) B= argminb ∑i=1bωi,t>T

In Eq. (11), the threshold value T represents the proportion of each Gaussian component to the background model, usually taken as 0.75.

Comparison of Several Algorithms

In order to better describe the advantages and disadvantages of several algorithms, this article analyzes the processing results of certain video sequences. Figures 1 and 2 show the processing results of the inter frame difference method and background subtraction method under global vision, respectively. Figure 1A shows the two frame difference results, and Fig. 1B shows the three frame difference results. Due to the slow speed of the object, there are too many identical parts in two frames, meaning the detection results only have a relatively thick double edge contour, which is generally larger than the actual object. It also leads to the problem of internal holes within the object. The three-frame difference method is achieved through using the logical AND operation on the results of the two-frame difference method. It solves the problem of double edge overlap and is able to detect the object’s finer contours. This also leads to a large number of object pixels being eliminated for the logical AND operation, resulting in contour blurring. Figures 2A and 2B show the processing results of the single Gaussian model and the mixed Gaussian model, respectively. The detection results of these two Gaussian models are relatively complete, clear, and accurate. They are better than that of the two-frame difference and three-frame difference. The foreground area of the mixed Gaussian model is much larger than the result of the single Gaussian model. The details should also be clearer, such as better facial details in the first image. The noise of the mixture model image is much higher than that of the single model. This also proves that the Gaussian mixture detects more details but sacrifices some of its noise resistance.

Figure 1 (A) Two frame differential result. (B) Three frame differential result.

Figure 2 (A) Single Gaussian model result. (B) Mixed Gaussian model result.

Edge Extraction

The Canny edge detection operator is a multi-level edge detection algorithm developed by John F. Canny in 1986 (Lee, Tang & Park, 2016). Canny’s goal was to find an optimal edge detection algorithm, which included the following steps: (1) Applying Gaussian filtering to smooth images and remove noise; (2) calculating image gradients to obtain possible edges; (3) applying non maximum suppression techniques to eliminate edge misdetection; (4) applying a dual threshold method to filter edge information; (5) using lag technology to track boundaries.

The parameters of the Canny edge detector mainly include dual thresholds T1 and T2 in hysteresis threshold processing, represented by vectors as T = [T1, T2]. The other parameter, sigma, is the standard deviation of the smoothing filter. The value of the threshold vector T mainly affects the cleanliness of edge contour detection, which is positively correlated (Faheem et al., 2023).

In general, the edge function’s automatic threshold does not have a good contour effect. Moreover, due to changes in the video environment, it is not possible to always use a fixed threshold. Here, the Otsu method is used to provide the threshold number for the Canny edge detector in order to achieve better edge effects. The sigma number has little impact on edge detection performance, so the default value of 1 is set here. The Otsu algorithm is based on the statistical features of grayscale image histograms (Ranjitha & Shreelakshmi, 2021). It divides the grayscale image into two parts: background and foreground. The pixel grayscale value in the background is lower, while the pixel grayscale value in the foreground is higher. The calculation process of this method is selected based on the inter variance. The larger the variance is, the closer it is to the optimal threshold. This article uses the Otsu algorithm to calculate the optimal threshold as a high threshold (TH), and half of the high threshold as a low threshold (TL), so that the threshold vector T can be automatically calculated. Sometimes the threshold is set proportionally according to the actual situation. The following pictures are a comparison between the automatic threshold of Canny edge detector and the threshold obtained by the Otsu method. The threshold of the image is shown in Table 1.

From the data in Table 1, we can see that the threshold selected by the Otsu method is always larger than the automatic threshold. The smaller the threshold, the more edge information and details are detected, and the noise resistance is worse. Figures 3A and 3B show the processing results of a bridge using the two methods discussed above, respectively. There are always more details in image (a) than in image (b), such as corrosion marks on bridge piers. More detailed information is correlated to a higher susceptibility to interference, and the Gaussian mixture model is particularly sensitive to noise. However, the Otsu threshold segmentation results in less detailed information, and the overall edge of the object is also segmented. Therefore, the optimal threshold extraction effect of the Otsu method is better than that of automatic threshold extraction.

Table 1 Canny threshold selection.

Automatic threshold	Otsu threshold	
[0.0250,0.0625]	[0.0533,0.1066]	

Figure 3 (A) Automatic threshold. (B) Otsu threshold.

Improved Mixed Gaussian Model

The comparison results in the third part show that although the mixed Gaussian model has a relatively accurate detection effect, in complex environments the detection results still contain a lot of noise, and the foreground contour is too large, which affects the subsequent recognition effect. Therefore, this article uses edge contours to improve the mixed Gaussian model and reduce the impact of these two issues on the results. Less noise interference can improve the recognition rate of detection. Filtering and denoising the original image directly not only requires a lot of computation, but also results in the loss of some edge information. Median filtering can effectively remove noise in images, and we used this method to denoise foreground images. This article proposes a mixed Gaussian model based on feature fusion to address the issues of unclear foreground contours and excessive noise points in mixed Gaussian models, as shown in Fig. 4. The algorithm steps are as follows:

Figure 4 Improved Mixed Gaussian Model Steps.

1. Extract each frame of the video as an RGB image. Then the image in the RGB color space is converted to Grayscale in the HSV color space (Sheng, Wu & Wang, 2019). The HSV color space is similar to the colors used in painting, and is closer to peoples’ subjective perception of colors compared to the RGB color space. 2. Use the Gaussian Mixture model to extract the foreground image to be used as the input image of the next step. 3. Use median filtering to remove a small amount of noise from the foreground image to obtain a pure foreground object area. Then dilate the foreground object area. 4. Use the first frame as the background frame of the Gaussian model. Then use the Canny edge detector to extract the contour of the image from the second frame and save it. 5. Perform logical operations on foreground target areas and edge images. Then, repeat steps (2) to (5) for each extracted image and save the results. 6. Output the previously processed image results.

Experimental Results Analysis

To verify the effectiveness of the algorithm proposed in this article, two videos were captured on a mobile phone for algorithm validation. The phone used was a MEIZU17 Pro, with a Sony 6-piece lens camera with video parameters of 1280 * 720p and 30fps. When shooting a video, the camera needs to be fixed and zoomed out. For processing convenience, the video was compressed to 480 × 360 pixels. The computer operating system was a standardized 64 bit Win10 with 16G RAM and a i7-10510U CPU. The software used to run the experimental code for this article was Matlab2019b, which does not need additional data preprocessing such as deep learning, and only uses the original image.

Figure 5 257th, 258th, 259th frames of Video S1.

Figure 6 328th, 329th, 330th frames of Video S2.

Figures 5 and 6 show several images from Videos S1 and S2, respectively. Among them, frame k is the background image, and frames k +1 and k +2 are the foreground detection images to be processed by the mixed Gaussian model. Through the algorithm steps defined in Section 5, the contour region of the k +2 frame is extracted, and the final results are shown in Figs. 7 and 8. Figure 7A shows the foreground objects extracted from tennis videos. Figure 7B shows the resulting image after median filtering and dilation operation. Figure 7C shows the edge contour of the canny algorithm improved by Otsu. Figure 7D shows the final foreground object contour after feature fusion. The results in Figs. 8A–8D represent the same processing steps as those in Figs. 7A–7D, with different videos and foreground objects. From the above results, the parameters of the mixed Gaussian model are shown in Table 2.

Figure 7 (A–D) Step Results for Video S1.

Figure 8 (A–D) Step Results for Video S2.

Table 2 Parameters of mixed Gaussian model.

parameters	Value	
Deviation threshold D	2.5	
Learning rate a	0.01	
Foreground threshold T	0.75	
Initialization standard deviation S	10	
Number of Gaussian models K	3	

Figure 9 Optical flow vector of detection results.

Figure 10 Object area after thresh comparison.

The results clearly show that after median filtering, almost all excess noise in the foreground image extracted by the Gaussian mixture model has been removed, as shown in Figs. 7B and 8B. After using the logical AND operation on Figs. 7B and 7C, the results shown in Fig. 7D can be obtained. From Fig. 7D, the edge contour of the moving object is clearly visible, and the detection effect is significantly improved. For a more comprehensive experiment, Video S3 is processed by the improved method in this article and then compared with the optical flow method. Video S3 comes from the third-party LASIESTA datasets with ID O_CL_02 (Cuevas, Yáñez & García, 2016). The databases are complete, useful, and publicly available (http://www.gti.ssr.upm.es/data/LASIESTA). The 120th frame in Video S3 is used as an example, and the processing effect on the female person is compared. Figure 9 shows the detection results of the optical flow method with the blue dots indicating the optical flow vector. The foreground object is obtained by threshold comparison, as shown in Fig. 10. There are still a large number of false candidate objects in the background which are caused by light changes and pixel noise. There are also a large number of holes inside the object. This is because the internal pixel movement is basically unchanged and the optical flow vector value is small. The boundary region is roughly detected, due to significant changes in the gradient of the optical flow vector of the object boundary pixels. Although the optical flow method can detect the boundary area, the boundary is coarse. It still cannot obtain accurate object contours. The GMM method can effectively detect the foreground object, with significantly more object pixels than the optical flow method. The amount of noise pixels in the foreground object are obviously fewer than in the optical flow method, and the object’s overall shape is essentially complete, as shown in Fig. 11. The detection results of the proposed method are shown in Fig. 12. Compared with the above two methods, this improved method has less foreground noise, better accuracy in the overall object contour area, and fewer internal defective pixels. It can provide complete pixel information for the next object recognition. However, there are still some discontinuous edge segments and edge defects in the final resulting image.

Figure 11 Foreground object after GMM.

Figure 12 Contour after improved GMM.

Conclusion

The problem of object detection in the global visual environment is an important subset of object detection issues. This article first analyzes the advantages and disadvantages of various methods, then focuses on the theoretical principles of single Gaussian models and Gaussian mixture models. Then, the problems associated with commonly used algorithms were explained through a comparative example, and the fusion idea was provided. Following an explanation of the principle of edge detection algorithms with automatic thresh, the improvement of the edge detection effect with thresh generated by the Otsu method was discussed. The object detection performance of the GMM model was improved through edge contours, and detailed steps of the algorithm were provided. Finally, through a comparison experiment, the performance of the proposed algorithm was verified. The experimental results show that the improved GMM model in this article enhances the accuracy of object detection contour and reduces the noise of the object. In terms of future research, it may be promising to address the problem of discontinuous edge segments to improve algorithm performance. Alternatively, the improved method could be combined with the convolutional neural networks to generate a new detection method.

Supplemental Information

Videos S1 & S2 Two videos from the experimental analysis section and images used in the paper main body

Click here for additional data file.

Video S3 Video 3 comes from the LASIESTA for added comparative experiment

Click here for additional data file.

Supplemental Information 3 Comparation algorithm code and Experimental results analysis code

Click here for additional data file.

Many thanks to professor Carlos Cuevas of Universidad Politécnica de Madrid for the contribution to the public LASIESTA dataset. The author is also grateful for the anonymous reviewers and editor who made constructive comments.

Additional Information and Declarations

Competing Interests

Author Contributions

Data Availability

The authors declare there are no competing interests.

Lei Sun conceived and designed the experiments, performed the experiments, analyzed the data, performed the computation work, prepared figures and/or tables, authored or reviewed drafts of the article, and approved the final draft.

The following information was supplied regarding data availability:

The raw data and code are available in the Supplemental Files.

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
