# Peer review of "Global vision object detection using an improved Gaussian Mixture model based on contour"

_PeerJ Computer Science, doi:10.7717/peerj-cs.1812_

## Round 0.1 · original submission · Minor Revisions

1. The paper lacks relevant important literature.
2. The paper lacks comparison with existing methods.
Please revise your manuscript carefully according to the reviewers' comments.

**Language Note:** The review process has identified that the English language must be improved. PeerJ can provide language editing services - please contact us at copyediting@peerj.com for pricing (be sure to provide your manuscript number and title). Alternatively, you should make your own arrangements to improve the language quality and provide details in your response letter. – PeerJ Staff

Reviewer 1 ·

Basic reporting

The writing is clear at most places and also it has given the comparison of several different algorithms in section3.So the new readers can understand why should we proposed the new method. The figures help quickly grasp the advantages or disadvantages in the paper.

Experimental design

comprehensive experiments to demonstrate the validity of the proposed technique.

Validity of the findings

The reviewer has gone through the proposed method in Section 5 and found that it is indeed practical.

Additional comments

Visual object detection has a variety of significant applications such as human-computer interfaces, road traffic control, and video surveillance systems. Aimed at the problem that the foreground contour of moving object detection is not clear and there are too many noise points in the global vision indoor environment, the author proposes an improved mixed Gaussian model. First of all, the reviewer enjoyed reading this article and appreciated the comparative experiments to support the proposed method.

The idea of the paper and the proposed solution is convincing. However, this paper still has some flaws; hence accept it after minor revisions. The suggestions are listed as follows:

Suggestions to Improve:
(1) The introduction should add several latest articles related to GMM
(2) The clarity of the images should be increased to make the paper more aesthetically pleasing.
(3) The formula “eq” in the paper should use corrective standard abbreviations
(4) The citation format of literature should be modified according to the requirements of the journal.

Questions for clarification:
(1) There are many classic methods for edge extraction, such as prewitt, sobel, etc. Why do you choose the Canny algorithm.
(2) Why not use RGB images for improved algorithms, but convert them into HSV space?

·

Basic reporting

This paper is nice and has mathematical real-world applications; I see many positive aspects in this work and would like to see it published. In fact, this paper will be of value and interest to as a significant portion of potential readers of the journal.

Experimental design

The authors should clarify if the proposed method requires data processing or not. More detail information should be added to make clear explanations.

Validity of the findings

1. Some parts of mathematic derivations are not given in details. I suggest the authors a carefully checking and give all the necessary manipulations for the derived formulas.

2. More detailed review of the literature is expected in separate section. Specially, it is required that the previous solutions to this problem be addressed. Then, the advantages (and disadvantages?) of the proposed methods and should be discussed.

Additional comments

1. The contributions should be more clearly explained with more details on how to improve the existing results, especially in the references the authors cited.

2. Some latest references about object detection should be added to give readers an up-to-date picture. In this sense, the following papers can be referred: Intelligent detection and behavior tracking under ammonia nitrogen stress, Neurocomputing; KD-PAR: a knowledge distillation-based pedestrian attribute recognition model with multi-label mixed feature learning network, Expert Systems With Applications; AA-WGAN: attention augmented wasserstein generative adversarial network with application to fundus retinal vessel segmentation, Computers in Biology and Medicine.

3. Some future directions can be discussed in the conclusion part.

4. The authors still need a careful check of English, formulas and format/style.

---

## Round 0.2 · accepted · Accept

Thank you for your improvement.

·

Basic reporting

no comment

Experimental design

no comment

Validity of the findings

no comment

Additional comments

no comment